# Deformable NeRF using Recursively Subdivided Tetrahedra

Zherui Qiu
University of Science and Technology
of China
Hefei, China
zrqiu@mail.ustc.edu.cn

Chenqu Ren
East China Normal University
Shanghai, China
51255903034@stu.ecnu.edu.cn

Kaiwen Song
University of Science and Technology
of China
Hefei, China
SA21001046@mail.ustc.edu.cn

Xiaoyi Zeng
University of Science and Technology
of China
Hefei, China
zxy1908542805@mail.ustc.edu.cn

Leyuan Yang
University of Science and Technology
of China
Hefei, China
ly_1207@mail.ustc.edu.cn

Juyong Zhang*
University of Science and Technology
of China
Hefei, China
juyong@ustc.edu.cn

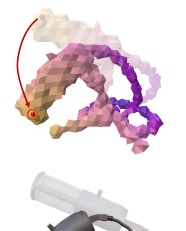 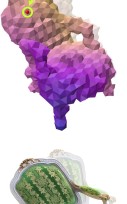 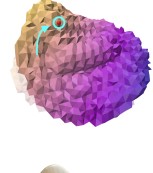 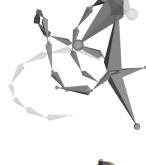 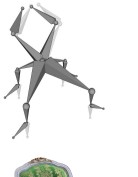 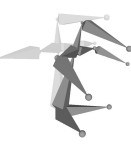
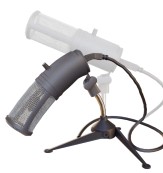 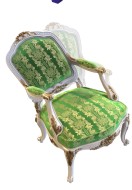 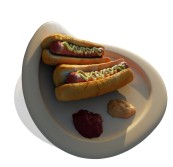 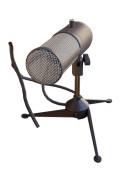 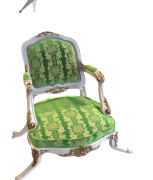 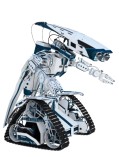

Figure 1: Illustration of our method's integration of tetrahedral mesh manipulability with high-quality feature grid rendering capabilities. The left three columns demonstrate controlled deformations enabled by our approach, allowing for user-directed modifications. The right three columns showcase the application of our method in rigged animation.

## Abstract

While neural radiance fields (NeRF) have shown promise in novel view synthesis, their implicit representation limits explicit control over object manipulation. Existing research has proposed the integration of explicit geometric proxies to enable deformation. However, these methods face two primary challenges: firstly, the time-consuming and computationally demanding tetrahedralization process; and secondly, handling complex or thin structures often leads to either excessive, storage-intensive tetrahedral meshes or poor-quality ones that impair deformation capabilities. To address these challenges, we propose DeformRF, a method that seamlessly integrates the manipulability of tetrahedral meshes with the high-quality rendering capabilities of feature grid representations. To avoid ill-shaped tetrahedra and tetrahedralization for each object, we propose a two-stage training strategy. Starting with an almost-regular tetrahedral grid, our model initially retains key tetrahedra surrounding the object and subsequently refines object details using finer-granularity mesh in the second stage. We also present the concept of recursively subdivided tetrahedra to create higher-resolution meshes implicitly. This enables multi-resolution encoding while only necessitating the storage of the coarse tetrahedral mesh generated in the first training stage. We conduct a comprehensive evaluation of our DeformRF on both synthetic and real-captured datasets. Both quantitative and qualitative results demonstrate the effectiveness of our method for novel view synthesis and deformation tasks. Project page: https://ustc3dv.github.io/DeformRF/

*Corresponding author

## CCS Concepts

• **Computing methodologies** → **Rendering**; *Animation*; **Computer vision**.

## Keywords

Neural Radiance Fields, Tetrahedral Mesh, Deformation

**ACM Reference Format:**
Zherui Qiu, Chenqu Ren, Kaiwen Song, Xiaoyi Zeng, Leyuan Yang, and Juyong Zhang. 2024. Deformable NeRF using Recursively Subdivided Tetrahedra. In *Proceedings of the 32nd ACM International Conference on Multimedia (MM '24), October 28–November 1, 2024, Melbourne, VIC, Australia.* ACM, New York, NY, USA, 9 pages. https://doi.org/10.1145/3664647.3681019

## 1 Introduction

The study of novel view synthesis has witnessed remarkable advancements in the fields of computer vision and graphics, notably since the inception of Neural Radiance Fields (NeRF) [26]. Subsequent to NeRF, numerous follow-up works have emerged, among which methods based on feature grids [4, 5, 22, 27, 31, 34] have demonstrated exceptional performance in terms of rendering quality and speed. Despite the promising results, the inherent nature of NeRF as an implicit continuous representation does not directly afford explicit control mechanisms. This limitation poses significant challenges in the deformation and manipulation of objects, especially in scenarios requiring user-directed control for precise adjustments and interventions.

To address this challenge, researchers have proposed integrating explicit geometric proxies to enable deformation. Some works have utilized triangular meshes for deformation, including the method proposed by Xu et al. [39], which generates watertight triangular meshes encapsulating the target object. Alternative studies [12, 23, 36, 40] have employed tetrahedral meshes as geometric proxies, building upon the feature grid approach. By manipulating the shape of these meshes, they facilitate the deformation of the object in a controlled manner. However, these methods face two main challenges: Firstly, they require a time-consuming and computationally intensive process of applying Marching Cubes algorithms [24] and tetrahedralization after training the implicit representation. Secondly, this process becomes particularly challenging for objects with complex topology or those containing thin structures, as it tends to result in one of two scenarios. In one scenario, to accurately conform to the object's intricate details, an excessive quantity of small tetrahedra is generated, significantly increasing the storage overhead. Alternatively, limiting the total number of tetrahedra often results in the creation of poor-quality tetrahedra, which adversely affects the efficacy of deformation tasks.

In this paper, we introduce a novel method, termed DeformRF, that seamlessly integrates the manipulability of tetrahedral meshes with the high-quality rendering capabilities of feature grid representations. Our method effectively circumvents the issues associated with poor-quality tetrahedra and also eliminates the need for tetrahedralization for each object. Departing from traditional practices, we propose a two-stage training framework, which offers a one-time solution for tetrahedralization and avoids the time-consuming Marching Cubes process for surface mesh extraction. Initially, we generate a regular, fixed tetrahedral grid before training, employing this uniform grid across all objects. During the first training stage, our model identifies and retains the crucial tetrahedra that effectively encapsulate the target object. Subsequently, the second training stage leverages the tetrahedral mesh obtained from the first stage. In this stage, the training of the tetrahedral representation is elevated by increasing the subdivision levels, thereby enhancing the model's visual fidelity. The two-stage training framework streamlines the training process and produces high-quality tetrahedra, which greatly benefits subsequent deformation tasks.

In order to fully harness the advantages of feature grid methods in rendering while effectively integrating with tetrahedral structures, it seems intuitive to store features at the vertices of the tetrahedra. Tetra-NeRF [16] has implemented this approach, leading to impressive rendering results; however, it also encounters its own set of challenges. Its tetrahedral mesh, derived from a dense point cloud through triangulation, leads to high memory usage and prolonged training times. To tackle these challenges, we introduce the concept of recursively subdivided tetrahedra, which virtually generates detailed meshes without the necessity to store these subdivisions. This approach enables multi-resolution feature encoding while only necessitating the storage of a coarse tetrahedral mesh generated during the first stage of training. From the initial coarse mesh, we obtain progressively finer levels of the tetrahedral mesh by repeating connecting the midpoints of each tetrahedron's edges. For a given sample point on the camera ray, we conduct a barycentric interpolation at each hierarchical level of the model, utilizing the features located at the vertices of the encompassing tetrahedron relevant to that level. The distinct feature vector from each level are subsequently concatenated to form a comprehensive feature vector for the sample point. We observe that the barycentric coordinates at a current level can then be used to calculate these coordinates at a subsequent, finer level of the hierarchy. To capitalize on the inherent hierarchical structure of recursively subdivided tetrahedra, we introduce an iterative algorithm for computing barycentric coordinates at each level of subdivision. This strategy significantly reduces memory demands by storing only the coarse tetrahedral mesh, thus avoiding the need to maintain vertex and connectivity information for the finer meshes.

We conduct a comprehensive evaluation of our DeformRF on both synthetic and real-captured datasets. Both quantitative and qualitative results demonstrate the effectiveness of our method for novel view synthesis and deformation. In summary, our work has the following contributions:

- We propose a novel approach that combines the manipulability of tetrahedral meshes with the high-quality rendering capabilities of feature grid representations.
- Our approach introduces an iterative computation of barycentric coordinates, eliminating the need for storing high resolution meshes explicitly. This not only lessens the memory footprint but also maintains the enhanced detail and accuracy of the finer tetrahedral mesh.
- The proposed method extends the capabilities of NeRFs to include explicit object-level deformations and animations while preserving photorealistic rendering quality.

## 2 Related Work

### 2.1 Neural Scene Representation

Significant advancements in novel view synthesis have been driven by NeRF [26], which employ multi-layer perceptrons (MLP) to effectively model both the geometric and appearance aspects of scenes, leveraging volume rendering to deliver high-quality renderings. NeRF has found extensive applications in versatile computer graphics and computer vision tasks, such as human avatar creation [8, 29], pose estimation [2, 6, 21], reconstruction [3, 33], robotics [1, 20] and simulation [18, 19]. However, despite these advancements, NeRF still contends with challenges in temporal and resource consumption efficiency. To mitigate the issues of time and memory inefficiencies, the training and representation capabilities of NeRF

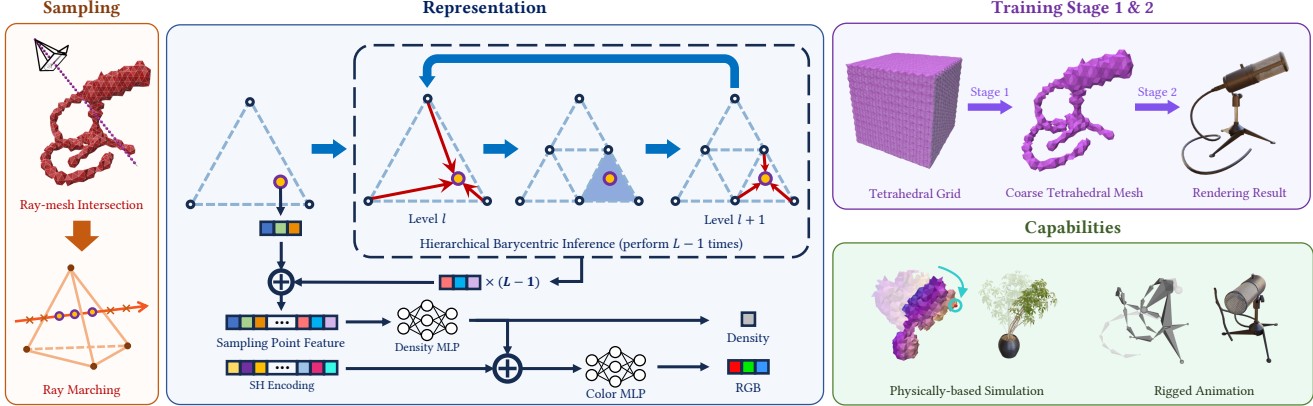

**Figure 2: Overview of DeformRF. (a) Given that a ray intersects with a tetrahedral mesh, we proceed with ray marching while retaining the sample points within the tetrahedron. (b) For each sample, we perform barycentric interpolation at each level and combine the feature vectors from all levels to create a complete feature vector. In this process, the computation of the barycentric coordinates is conducted iteratively. (c) In the two-stage training process, we first acquire a coarse mesh and then enhance training through increased subdivisions. (d) Our method support physically-based simulation and rigged animation.**

have been improved through the use of feature-grid based scene representations [4, 5, 22, 27, 31, 34].

In addition to methods based on regular grids, some approaches utilizing tetrahedral grids have also demonstrated impressive rendering capabilities. Tetra-NeRF [16] uniquely represents scenes with tetrahedral meshes, derived from Delaunay triangulation of a point cloud, and these meshes are then used for efficient barycentric interpolation of features, with a shallow MLP predicting density and color for volume rendering. However, the mesh generated by Delaunay triangulation is not closely adhered to object surfaces and contains many ill-shaped tetrahedra, making it unsuitable for deformation. PermutoSDF [30] combines permutohedral lattice hash encoding with a neural implicit surface model, integrating regularization techniques for smooth surface recovery and precise geometric reconstruction. Nevertheless, the relationship between grids of different levels in PermutoSDF is not hierarchical, which limits its capacity for deformation. Lately, 3D Gaussian Splatting [13] has gained prominence, showcasing notable real-time results in novel view synthesis as well.

However, unlike our work, these studies focus solely on achieving high-quality rendering in static scenes and do not explore the potential for extending their techniques to encompass manipulation of the objects. In contrast, our method not only capitalizes on the high-quality rendering advantages of feature grid-based approaches, but also supports deformation and animation.

## 2.2 3D Deformation and Manipulation

In the context of computer graphics, manipulating a 3D model refers to the process of altering its geometry based on user-defined parameters or controls. PIE-NeRF [7] employs quadratic generalized moving least squares to achieve meshless discretization, integrating physics-based simulations with NeRF. However, this approach does not directly support rigging, a common technique used in character animation and deformation that relies on a predefined

skeletal structure to control the deformation of a mesh. Xu et al. [39] proposed a method for free-form deformation of radiance fields using cage-based deformation, which involves manipulating a coarse triangular mesh. However, the topological structure of triangular meshes may limit the types and extents of possible deformations.

Several studies [12, 23, 36, 40] have utilized advanced tetrahedral meshing algorithms [11, 32] to construct geometric proxies. In particular, NeRFshop [12] employs a region-growing algorithm to expand user selections into a larger volume, which is then converted into a surface mesh using Marching Cubes; this mesh is tetrahedralized with TetGen [32] to create a geometric proxy for deformation. These methods are effective for interior tetrahedralization of pre-existing, watertight surfaces. Nonetheless, these sophisticated techniques can be time-consuming and require significant computational resources.

3D Gaussian Splatting (GS) [13, 38], as an explicit discrete representation, facilitates more direct and straightforward editability. Despite this, it faces issues under large deformations, necessitating intricate algorithms to manage scale, rotation, translation, and the complex merging and splitting of Gaussian ellipsoids, often resulting in artifacts [9].

In contrast to the aforementioned methods, our work presents a streamlined approach to 3D object manipulation that combines the strengths of tetrahedral meshing with the high-fidelity rendering of feature-grid representation. By introducing a two-stage training framework, we bypass the need for complex and resource-intensive tetrahedralization processes. Furthermore, we introduce the concept of recursively subdivided tetrahedra to implicitly create higher-resolution meshes. This method enables efficient multi-resolution feature encoding and reduces memory usage while also introducing a hierarchical structure to tetrahedral meshes, specifically designed to support deformation and animation.

# 3 Method

In this paper, we aim to seamlessly integrate the manipulability of tetrahedral meshes with the high-quality rendering capabilities of feature grid representations. Fig. 2 highlights how our DeformRF effectively combines tetrahedral mesh flexibility with advanced rendering capabilities, progressing from sampling and neural representation through sequential training stages to its diverse applications in simulation and animation.

In the following sections, Sec. 3.1 offers the necessary background for NeRF and multi-resolution hash encoding. Sec. 3.2 delves into our novel representation, which uses recursively subdivided tetrahedra to simulate increasingly refined and detailed meshes without actually storing them. This implicit subdivision method allows for detailed representations while conserving storage space and ensuring computational efficiency. Sec. 3.3 discusses the two-stage training process implemented in our model.

## 3.1 Preliminary

*3.1.1 Neural Radiance Fields.* In our approach, we utilize a differentiable volume rendering model akin to that employed in NeRF [26]. This model determines the color of a ray by integrating over contributions from points sampled along the ray path. The approximation is described by the equation:

$$\hat{C}(\mathbf{r}) = \sum_{i=1}^{N} T_i \left(1 - \exp(-\sigma_i \delta_i)\right) \mathbf{c}_i, \tag{1}$$

where

$$T_i = \exp\left(-\sum_{j=1}^{i-1} \sigma_j \delta_j\right). \tag{2}$$

In these expressions, $T_i$ denotes the light transmittance through ray $\mathbf{r}$ up to the $i$-th sample, effectively capturing the light contribution from all preceding samples. The term $(1 - \exp(-\sigma_i \delta_i))$ represents the contribution from the $i$-th sample itself, with $\sigma_i$ being the opacity of the sample, and $\delta_i$ the distance to the next sample along the ray. The variable $\mathbf{c}_i$ specifies the color at the $i$-th sample.

*3.1.2 Multi-resolution Hash Encoding.* Instant-NGP [27] introduced an encoding strategy for neural graphics primitives. Specifically, for a given point $\mathbf{x}$, hash encoding maps features from a cascade of grids at each level through a spatial hash function [35] :

$$h(\mathbf{x}) = \left(\bigoplus_{i=1}^{d} x_i \pi_i\right) \bmod T_h, \tag{3}$$

where $\oplus$ denotes the bit-wise XOR operation , $\pi_i$ represents a unique large prime number associated with each dimension and $T_h$ is the size of hash table. These features are then interpolated using trilinear interpolation, and features from multiple levels are concatenated to determine the features for that point. Since the features are stored as trainable parameters, the size of the MLPs can be significantly reduced, thereby saving both training and rendering times.

## 3.2 Multi-Resolution Tetrahedral Representation

In this subsection, we introduce two key components of our method: a multi-resolution representation based on recursively subdivided

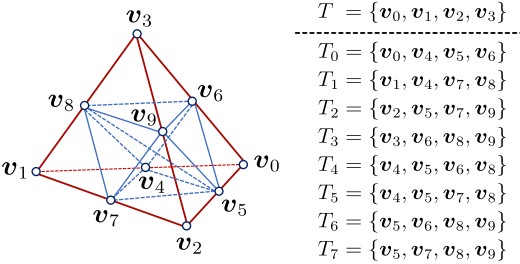

**Figure 3: Subdivision of a Tetrahedron. Given a tetrahedron $T$, we subdivide it into eight smaller tetrahedra $T_k$, for $k \in \{0, \ldots, 7\}$, by connecting the midpoints of each edge.**

tetrahedra and an efficient approach for iterative barycentric coordinates computation. Here we define $L$ as the total number of levels in the subdivision hierarchy, which includes the initial coarse mesh level and the total number of subdivision levels.

*3.2.1 Feature Encoding.* Inspired by the efficient encoding strategy of Instant-NGP [27], our method innovatively adapts this technique to the multi-resolution tetrahedral meshes. For each level of the tetrahedral mesh, the vertices are linked to the entries within their respective hash table. We use the same spatial hash function as Instant-NGP [27] to map a vertex position $\mathbf{x}$ to its feature vector. Unlike previous feature-grid methods, which employ regular grids, our approach uses tetrahedral meshes, necessitating the implementation of barycentric interpolation for feature interpolation within these meshes. We use the barycentric coordinates of each sampling point to calculate a weighted sum of the vertex features of the tetrahedron that contains the point. This process effectively yields the feature of the sampling point at each level. Subsequently, these features, derived from different levels of the tetrahedral mesh, are concatenated to form a comprehensive feature vector for the sample point.

*3.2.2 Recursively Subdivided Tetrahedra.* In our approach, we employ an intuitive method to generate tetrahedral meshes of different levels from a coarse mesh, and infer these higher-resolution meshes on the fly without storing them. Given a coarse tetrahedral mesh, we execute a recursive subdivision process to generate tetrahedral meshes of increasingly higher resolution across multiple levels. Starting with the coarse tetrahedral mesh, each tetrahedron is subdivided into eight smaller tetrahedra by connecting the midpoints of its six edges. Note that there are multiple ways to connect the midpoints on edges. Here we select a specific subdivision pattern and uniformly apply this pattern to all tetrahedra in the mesh, as shown in Fig. 3. This subdivision process is recursively applied, transforming each tetrahedron at level $\ell$ into eight smaller tetrahedra at level $\ell + 1$, which leads to progressively finer tetrahedral meshes at each level.

Although higher resolution meshes enable a more detailed representation of objects, they correspondingly demand greater memory space. It is impractical to store all meshes in memory simultaneously. We observe that storing high-resolution meshes is not necessary for our multi-resolution tetrahedral representation, as the essential information for feature encoding can be obtained by concentrating

on two key aspects. Firstly, identifying the coordinates of the four vertices of the tetrahedron at the sampling point allows us to access the features of these vertices. Secondly, calculating the barycentric coordinates at the sampling point within the tetrahedron is crucial for enabling barycentric interpolation. Once these two key aspects are addressed, the process of feature encoding can be successfully completed.

*3.2.3 Hierarchical Barycentric Inference.* Building on the need to encode features through vertex identification and barycentric calculations, we propose an iterative approach that simplifies the process based on the hierarchical structure of our recursive subdivided tetrahedra. This method allows us to determine which of the eight child tetrahedra at level $\ell + 1$ contains a specific sampling point at level $\ell$, using the barycentric coordinates of the point in the parent tetrahedron. Moreover, the barycentric coordinates of the sampling point at level $\ell + 1$ can be explicitly calculated from its coordinates at level $\ell$, using a closed-form formula.

Initially, our method employs the NVIDIA OptiX library [28] to perform ray-tetrahedral mesh intersections. Subsequently, we apply ray marching along the camera rays to generate a series of sampling points. Utilizing the results of these intersections, along with the positions of the sampling points, we are able to determine within which tetrahedron of the coarse mesh each point resides, as well as compute the corresponding barycentric coordinates.

Having obtained the barycentric coordinates for sampling points at the coarsest level, our focus shifts to identifying which of the eight child tetrahedra each sampling point falls into. Without loss of generality, consider an arbitrary tetrahedron in level $\ell$ defined by its four vertices $\mathbf{v}_0^\ell, \mathbf{v}_1^\ell, \mathbf{v}_2^\ell, \mathbf{v}_3^\ell \in \mathbb{R}^3$. For a given sampling point $\mathbf{p}$ located within this tetrahedron, assume its barycentric coordinate in level $\ell$ is known and denoted as $\boldsymbol{\alpha}^\ell = (\alpha_0^\ell, \alpha_1^\ell, \alpha_2^\ell, \alpha_3^\ell)$. The coordinates of $\mathbf{p}$ can thus be expressed as:

$$\mathbf{p} = \alpha_0^\ell \mathbf{v}_0^\ell + \alpha_1^\ell \mathbf{v}_1^\ell + \alpha_2^\ell \mathbf{v}_2^\ell + \alpha_3^\ell \mathbf{v}_3^\ell. \tag{4}$$

Upon subdividing the original tetrahedron by connecting these midpoints as shown in Fig. 3, eight smaller tetrahedra are formed. The sampling point $\mathbf{p}$ will be contained within one of these eight tetrahedra, and there exist eight possible configurations for its placement.

To determine the specific child tetrahedron containing the sampling point $\mathbf{p}$, we apply a set of criteria based on the barycentric coordinates. If any of the $\boldsymbol{\alpha}^\ell$ values, such as $\alpha_i^\ell$, exceeds 0.5, the sampling point is located at one of the four corners of the parent tetrahedron; otherwise, if all values are less than 0.5, it is positioned within one of the central four child tetrahedra. In cases where any of $\alpha_i^\ell$ equals 0.5, the sampling point theoretically lies on the shared face between two adjacent child tetrahedra. In our approach, such a point can be assigned to either of the adjoining tetrahedra without affecting the outcome. If the sampling points are not located at the corners, we then narrow down the child tetrahedron's identification by checking if the sums $\alpha_1^\ell + \alpha_2^\ell$ and $\alpha_2^\ell + \alpha_3^\ell$ exceed 0.5. Depending on whether these sums exceed 0.5, we can accurately pinpoint the child tetrahedron that contains $\mathbf{p}$. The detailed criteria are provided in the supplementary material. After determining the specific child tetrahedron containing the sampling point, we are able to access

the positions of the child tetrahedron's vertices. This enables us to retrieve the features of these vertices at level $\ell + 1$.

Recall that the barycentric coordinates of the sampling point at level $\ell + 1$ are required to perform barycentric interpolation. Let the barycentric coordinates of $\mathbf{p}$ in the specific child tetrahedron be $\boldsymbol{\alpha}^{\ell+1} = (\alpha_0^{\ell+1}, \alpha_1^{\ell+1}, \alpha_2^{\ell+1}, \alpha_3^{\ell+1})$. For each of the eight child configurations resulting from the subdivision of a parent tetrahedron, a distinct coefficient matrix $\mathbf{C}_i \in \mathbb{R}^{4 \times 4}$, where $i = 0, \ldots, 7$, can be defined such that $\boldsymbol{\alpha}^{\ell+1} = \mathbf{C}_i \boldsymbol{\alpha}^\ell$. Since our subdivision pattern is predetermined and uniformly applied across all tetrahedra, as detailed in Fig. 3 and previous discussions, each coefficient matrix $\mathbf{C}_i$ is constant and specific to its configuration. These constant matrices enable the precise determination of the point's coordinates in the subdivided meshes at each successive level. Consequently, the barycentric coordinates at level $\ell + 1$ can be calculated from those at level $\ell$ using the appropriate $\mathbf{C}_i$, ensuring accurate and efficient interpolation across all levels of the mesh hierarchy. The detailed computation process, including the derivation of the coefficient matrix $\mathbf{C}$ that relates the barycentric coordinates from one level to the next, is elaborated in the supplementary material.

## 3.3 Two-stage Training

To reduce memory usage on the GPU and enhance training speed of our model, we propose a two-stage training process.

In the first stage, we employ the QuarTet algorithm [17] to generate the initial tetrahedralization of the space, resulting in a tetrahedral mesh that fills the space of a unit cube. We use an occupancy mesh to determine which tetrahedra to retain within the mesh, which is similar to how Instant-NGP [27] uses an occupancy grid to decide voxel retention. At the end of this phase, we obtain a coarse tetrahedral mesh that roughly envelops the object. The advantage of this is the rapid construction of an initial structure surrounding complex geometries while minimizing memory consumption by retaining only those tetrahedra that are actually occupied.

In the second stage, we elevate the tetrahedral representation training by increasing subdivision levels $L$. This phase is dedicated to refining the tetrahedral representation obtained from the first stage, aiming to achieve a more accurate and visually appealing neural representation. This refinement process demands more computational resources, but since it builds upon an already roughly formed structure, it allows for focused optimization of areas requiring detailed representation.

## 4 Experiments
## 4.1 Implementation Details

*Tetrahedral Mesh Initialization.* For constructing our initial tetrahedral mesh, we employ the QuarTet algorithm [17] for converting surface geometry into tetrahedral meshes. We input a cube with an edge length of 1 and set the grid spacing parameter to 0.02, which results in the generation of a tetrahedral mesh comprising 17,933 vertices and 92,234 tetrahedra.

*Experiment Setting.* We run our experiments on a workstation with an Intel Xeon E5-2690 v3, an NVIDIA GeForce RTX 3090, and 128GB of RAM. We use Adam optimizer [14] with parameters $\beta_1 = 0.9, \beta_2 = 0.999, \epsilon = 10^{-15}$. We set the learning rate to $1 \times 10^{-2}$

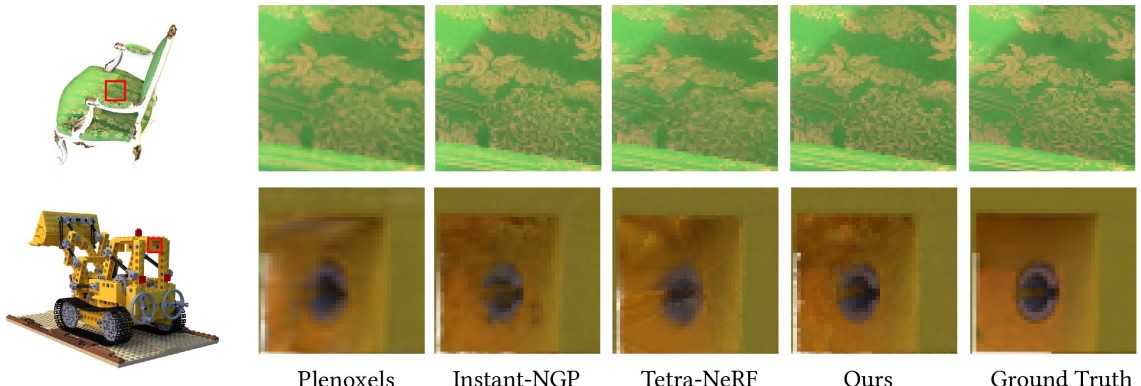

Plenoxels     Instant-NGP     Tetra-NeRF     Ours     Ground Truth

Figure 4: Qualitative Comparisons on the Synthetic NeRF Dataset [26].

for all scenes. Additionally, we apply a learning rate scheduler that gradually decreases the learning rate during training using a cosine annealing schedule. Furthermore, for our proposed method, we set the total number of levels in the subdivision hierarchy to 6 for all scenes.

## 4.2 Novel View Synthesis

To assess the effectiveness of our approach, we evaluate our model on both synthetic and real-captured datasets. The evaluation focuses on the fidelity of view synthesis in comparison to ground truth images captured from identical poses.

*Metrics.* This evaluation employs three distinct metrics: Peak Signal-to-Noise Ratio (PSNR), Structural Similarity Index Measure (SSIM) [37], and Learned Perceptual Image Patch Similarity (LPIPS) [41]. For Tetra-NeRF [16], we recalculated their SSIM metrics using the same SSIM implementation employed in the other methods for consistency.

*Datasets.* The Synthetic NeRF dataset [26] comprises eight intricately designed synthetic objects. Each object is captured through 100 images from virtual cameras strategically placed around a hemisphere facing inwards. Mirroring the methodology in NeRF [26], we utilize 100 views per scene for training purposes, reserving an additional set of 200 images for testing. The Tanks and Temples dataset comprises scenes of five real-world objects captured by an inward-facing camera orbiting the scenes. Each scene consists of 152 to 384 images with a resolution of $1920 \times 1080$.

*Quantitative Comparison.* We quantitatively compare various state-of-the-art methods on the Synthetic NeRF dataset and the Tanks and Temples dataset. Table 1 details the average performance across the metrics of PSNR, SSIM, and LPIPS for scenes from these datasets. The results from the Synthetic NeRF dataset demonstrate that our method surpasses other state-of-the-art methods in PSNR and SSIM. For the LPIPS metric, our method ranks second, just behind the NSVF method [22]. In the Tanks and Temples dataset, our method leads in both the PSNR and LPIPS metrics, while it holds the third position in SSIM, only surpassed by Plenoxels [31] and Tetra-NeRF [16].

Table 1: Qualitative Comparisons on the Synthetic NeRF Dataset [26] and the Tanks and Temples Dataset [15]. Best 3 scores in each metric are marked with gold ●, silver ● and bronze ●.

| Methods | Synthetic NeRF dataset | | | Tanks and Temples dataset | | |
|---|---|---|---|---|---|---|
| | PSNR ↑ | SSIM ↑ | LPIPS ↓ | PSNR ↑ | SSIM ↑ | LPIPS ↓ |
| NeRF [26] | 31.01 | 0.947 | 0.081 | 25.78 | 0.864 | 0.198 |
| NSVF [22] | 31.75 | 0.954 | 0.048 ● | 28.48 | 0.901 | 0.155 |
| Plenoxels [31] | 31.71 | 0.958 ● | 0.050 ● | 27.46 | 0.905 ● | 0.162 |
| Instant-NGP [27] | 32.68 ● | 0.948 | 0.054 | 28.62 ● | 0.890 | 0.142 ● |
| Tetra-NeRF [16] | 32.76 ● | 0.957 ● | 0.051 | 28.83 ● | 0.925 ● | 0.125 ● |
| Ours | 33.12 ● | 0.960 ● | 0.049 ● | 29.09 ● | 0.903 ● | 0.124 ● |

*Qualitative Comparison.* We selected several images with intricate details from the dataset to showcase the superior rendering quality of our method. Fig. 4 illustrates the performance of our method compared to strong baselines on the Synthetic NeRF dataset, where our approach more effectively captures the fine textures of objects and renders images with geometrically plausible appearances. Fig. 5 displays the comparative performance of our method against others on real-world data from the Tanks and Temples dataset. Our method achieves superior rendering results, particularly noticeable on surfaces with textual textures.

*Memory Usage.* To empirically validate the memory efficiency of our proposed method, we conducted a series of experiments comparing the memory usage of our approach with the Tetra-NeRF method. The experimental results are summarized in Table 2. Firstly, our approach achieves superior storage efficiency for tetrahedral meshes, as it only requires storing a coarse tetrahedral mesh, resulting in a much smaller storage size compared to Tetra-NeRF [16]. Secondly, our method shows considerably lower GPU memory consumption during training. Moreover, we investigated the differences in memory usage between single-stage and two-stage training. The results indicate that two-stage training is more memory-efficient, proving its effectiveness.

*Accumulation Map.* Since both our method and Tetra-NeRF [16] utilize tetrahedral mesh rendering, we conduct a comparative analysis from the perspective of accumulation to highlight differences

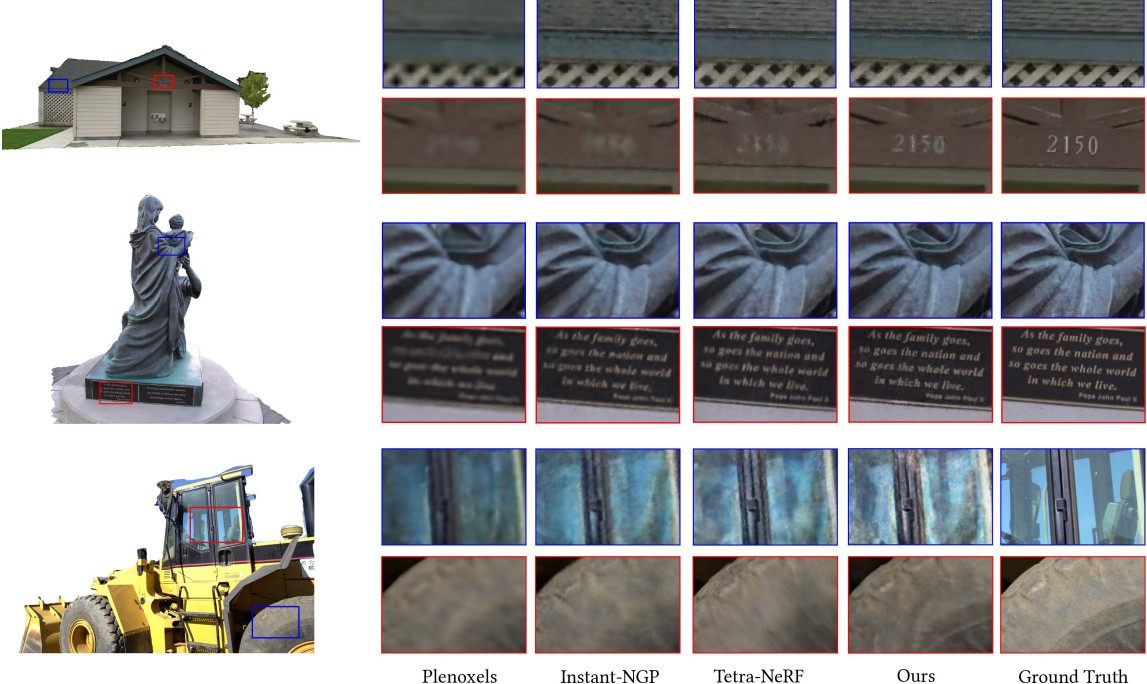

Figure 5: Qualitative Comparisons on the Tanks and Temples Dataset [15].

Table 2: Memory Usage Difference.

| | Synthetic NeRF dataset | | Tanks and Temples dataset | |
|---|---|---|---|---|
| | Ours | Tetra-NeRF | Ours | Tetra-NeRF |
| Mesh Storage Size | 248.88 KB | 1.25 MB | 677.20 KB | 110.85 MB |
| Stage 1 Memory | 5.34 GB | 18.39 GB | 7.64 GB | 17.75 GB |
| Stage 2 Memory | 2.70 GB | | 6.06 GB | |
| Single-stage Memory | 9.79 GB | - | 12.01 GB | - |

in performance. We generate accumulation images that represent the cumulative weighting of sampled points along a light ray's path. Each pixel in these maps is assigned a float value between 0 and 1, which reflects the extent of light interaction along the ray's path. While Tetra-NeRF suffers from ill-shaped tetrahedra within the tetrahedral mesh, frequently leading to artifacts in its accumulation maps, our method exhibits fewer artifacts as shown in Fig. 6. This suggests that our method avoids these mesh-related issues and achieves a more precise prediction and depiction of the densities within the scene.

## 4.3 Deformation

*4.3.1 Physically-based Simulation.* The coarse tetrahedral mesh serves as the geometric proxy for the simulation, offering a balance between computational efficiency and the ability to accurately represent complex deformations. The choice of a coarse mesh aids in reducing computational overhead while retaining sufficient detail to effectively capture the dynamics of the system. Upon establishing the tetrahedral mesh, we configure it as a mass-spring system. Each vertex of the mesh is treated as a mass point, with edges acting as

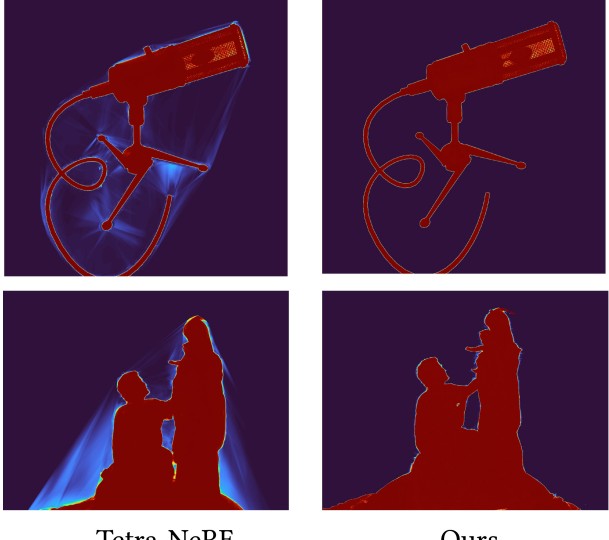

Tetra-NeRF                    Ours

Figure 6: Predicted Accumulation Map Comparisons on the Synthetic NeRF Dataset [26] and the Tanks and Temples Dataset [15].

springs that connect these mass points. To simulate the physics of our mass-spring system, we implement XPBD [25] using the Taichi programming language [10]. Particularly, we enforce constraints to maintain the invariance of spring length and tetrahedral volume.

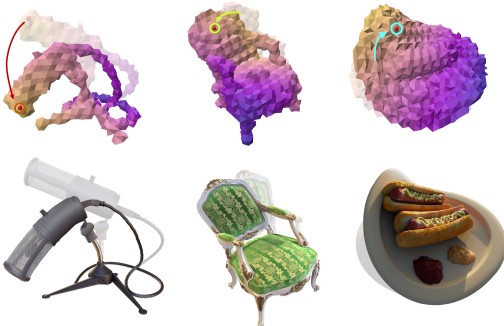

**Figure 7: Rendering Results of Simulation**

Fig. 7 displays the rendering results of an object deformed through user manipulation. The red spheres in the image represent the tetrahedral vertices dragged by the user's mouse. It is evident from the figure that our method maintains high-quality and photorealistic rendering even when the object undergoes deformation.

*4.3.2 Rigged Animation.* We first export the coarse tetrahedral mesh generated by our model into a format compatible with Blender. Utilizing Blender's robust rigging system, we attach a skeletal structure to the imported mesh. The process involves placing bones strategically within the mesh and ensuring that the weights assigned to each vertex were optimized for realistic deformation.

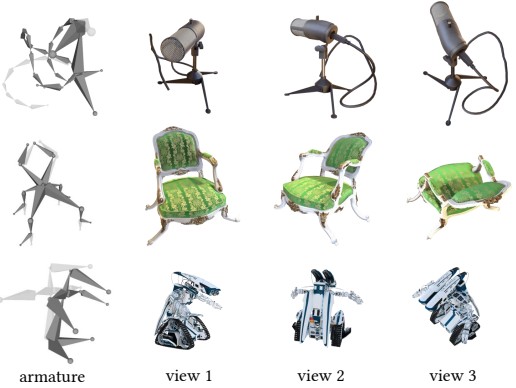

armature     view 1     view 2     view 3

**Figure 8: Rendering Results of Rigged Animation**

The rendering results of the rigged animations are presented in Fig. 8. Each row in the figure displays the armature along with rendering results from three different viewpoints. Our method successfully supports animations driven by skeletal structures, and under the designed poses, the rendering remains photorealistic.

Despite the complex deformations involved in the rigged animations, our model successfully maintains high-quality rendering, showcasing its robustness in handling dynamic changes.

## 4.4 Ablation Study

*4.4.1 Number of Subdivision Levels.* In our multi-resolution tetrahedral representation, the level of subdivision for the tetrahedra

is adjustable. We investigate the variations in the model's PSNR on the test set under different numbers of subdivision levels, as illustrated in Fig. 9. As expected, there is an improvement in performance with an increase in the number of levels. This demonstrates the efficacy of our subdivision strategy.

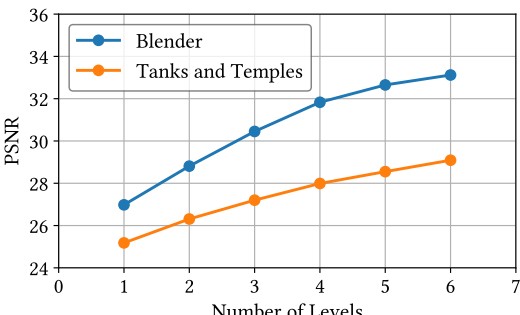

**Figure 9: Ablation Study of Number of Levels**

*4.4.2 Necessity of Two-stage Training.* To validate the effectiveness of our two-stage training strategy, we compare its rendering performance on the Tanks and Temples dataset [15] against that of a single-stage training approach. The single-stage training refers to direct model training using a full tetrahedral grid, instead of the process of generating a coarse tetrahedral mesh.

**Table 3: Quantitative Comparison between Single-stage Training and Two-stage Training.**

|  | PSNR↑ | SSIM↑ | LPIPS ↓ |
|---|---|---|---|
| Singe Stage | 28.01 | 0.887 | 0.155 |
| Two Stages | 29.09 | 0.903 | 0.124 |

We compare the average performance across the metrics of PSNR, SSIM, and LPIPS for scenes, as detailed in Table 3, where the two-stage training strategy exhibits a clear numerical advantage. This substantiates the necessity of employing a two-stage training strategy.

## 5 Conclusion

In this work, our DeformRF successfully integrates the manipulability of tetrahedral meshes with the high-quality rendering capabilities of feature grid representations. We have introduced the concept of recursively subdivided tetrahedra, which enables multi-resolution feature encoding by implicitly generating detailed meshes without the need to store each subdivision. By innovatively employing iterative computation of barycentric coordinates, we managed to maintain computational efficiency while dealing with higher mesh resolutions. Our comprehensive evaluations on both synthetic and real-captured datasets demonstrate the effectiveness of our method in novel view synthesis and deformation tasks. DeformRF advances the capabilities of neural radiance fields by enabling object-level deformations and animations while maintaining photorealistic rendering quality.

## Acknowledgments

This research was supported by the National Natural Science Foundation of China (No.62122071, No.62272433), and the Fundamental Research Funds for the Central Universities (No. WK3470000021). This research was also supported by the advanced computing resources provided by the Supercomputing Center of University of Science and Technology of China.

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
