# OpenReview forum: "Deformable NeRF using Recursively Subdivided Tetrahedra"
_acmmm.org/ACMMM/2024/Conference — MM2024 Poster_

### Official Review · Reviewer_dZNA · 2024-05-17

**Rating:** 3
**Confidence:** 3

**Summary:**

This paper combines tetrahedral mesh manipulability with feature grid rendering. A two-stage training strategy begins with a regular tetrahedral grid, refining details in a second stage with finer mesh. This approach uses recursively subdivided tetrahedra for multi-resolution encoding without excessive storage. Evaluations on synthetic and real datasets show its effectiveness in view synthesis and deformation tasks.

**Strengths:**

1. The paper targets a very interesting problem (the Manipulable NeRF) and proposes a promising method (NeRF on tetrahedral meshes) to solve it.
2. The paper achieves higher render image quality than some popular methods.

**Limitations:**

1. The contributions of this paper are not so clear. Please present a more detailed comparison with Tetra-NeRF. It also represents scenes with tetrahedral meshes. From my point of view, it should also support deformation and manipulation if these two abilities benefit from the tetrahedral meshes.
2. The analysis of inference efficiency is missing. I wonder whether this method can give real-time rendering since it requires multi-resolution subdivisions to obtain features. The Instant-NGP is not only accurate but also very fast. I suggest the authors give an inference efficiency comparison between their method and other methods, such as Instant-NGP and Tetra-NeRF.
3. While the authors say they try to tackle the challenges of high memory usage and prolonged training times in line 148, the paper does not present any experiment about memory consumption and storage efficiency. I suggest the authors provide detailed memory consumption and storage efficiency concerning different numbers of subdivision levels. And please quantitatively show that your method is more efficient than other methods.

**Suitability:**

2

---

### Official Review · Reviewer_HeNb · 2024-05-24

**Rating:** 4
**Confidence:** 3

**Summary:**

This paper proposes a manipulable NeRF using recursively subdivided tetrahedra. The recursively subdivided tetrahedra provides hierarchical features, enabling the high-quality rendering effect of NeRF. Furthermore, the recursively subdivided tetrahedra uses less memory than existing methods.
Finally, the paper achieves NeRF deformation, e.g., physically based simulation and rigged animation, by utilizing the manipulability of tetrahedral meshes. Comprehensive experiments demonstrate its effectiveness for novel view synthesis and deformation.

**Strengths:**

1. The paper is easy to read. The structure of the article is clear, and the figures are simple and easy to understand.
2. Combining recursive design with tetrahedra is impressive. The method significantly reduces memory consumption while enhancing rendering performance.

**Limitations:**

1. **Title**: Manipulation is a broad term that encompasses deformation and editing. This paper primarily achieves deformation. So, I think that "Deformable NeRF" more accurately describes the contribution of this paper than "Manipulable NeRF".

2. **Memory footprint**: The introduction mentions that recursively subdivided tetrahedra consume less memory, but no specific experiments are provided as evidence for this claim.

**Suitability:**

3

---

### Official Review · Reviewer_VJja · 2024-05-25

**Rating:** 3
**Confidence:** 1

**Summary:**

This work proposes Manipulable NeRF, a method that seamlessly integrates the manipulability of tetrahedral meshes with the high-quality rendering capabilities of feature grid representations. To avoid ill-shaped tetrahedra and tetrahedralization for each object, they propose a two-stage training strategy. Both quantitative and qualitative results demonstrate the effectiveness of our method for novel view synthesis and deformation tasks.

**Strengths:**

1. They propose a novel approach that combines the manipulability of tetrahedral meshes with the high-quality rendering
capabilities of feature grid representations.

2. They introduce an iterative computation of barycentric coordinates, eliminating the need for storing high-resolution
meshes explicitly. This not only lessens the memory footprint but also maintains the enhanced detail and accuracy
of the finer tetrahedral mesh.

**Limitations:**

1. Multi scale information fusion is a common technique in other fields to achieve better results, so is the first use of multi-resolution in tetrahedra pioneered in this article?

2. If the two stages are unified into one stage for training, will better results be obtained? The work indicates that this was considered to save memory, so how much memory does it differ from the two-stage training method?

In addition, I am not very familiar with this field, so I will revise my final score based on the author's response and the opinions of other reviewers

**Suitability:**

3

---

### Official Review · Reviewer_Y7Q3 · 2024-05-25

**Rating:** 3
**Confidence:** 3

**Summary:**

The paper introduces Manipulable NeRF, a method designed to enhance object manipulation in neural radiance fields while maintaining high-quality rendering. It employs a two-stage training strategy: starting with an almost-regular tetrahedral grid, it retains key tetrahedra and refines object details in subsequent stages. The method uses recursively subdivided tetrahedra for multi-resolution encoding, requiring only the storage of an initial coarse mesh.

**Strengths:**

1. The paper propose a novel approach that combines the manipulability of tetrahedral meshes with the high-quality rendering capabilities of feature grid representations.
2. The quantitative and qualitative results of the model performed very well.
3. The paper is clearly written and figures are useful.

**Limitations:**

1. The absence of time cost evaluation in the paper raises concerns about the practicality of the proposed Manipulable NeRF method. If the training time of Manipulable NeRF exceeds the time required to directly Tetrahedralize the results of existing state-of-the-art methods like instant-ngp or instant-neus[1], the significance of the paper is diminished.
2. The baseline in the experimental section of the paper lacks a comparison with the current mainstream 3D-GS series methods.
3. The novelty of the paper is insufficient. The main innovation is the use of Tetrahedra representation to achieve control of NeRF, but this control is a property inherent to the Tetrahedra representation itself, the idea of Recursively Subdivided seems to be quite common in graphics[21].

[1] https://github.com/bennyguo/instant-nsr-pl

**Suitability:**

3

---

### Meta-Review · Area_Chair_Uu4L · 2024-06-27

**Recommendation:** Accept (Poster)
**Confidence:** 4

**Metareview:**

This paper received a mixed score of weak accept and borderline. As this paper targets at an interesting problem of manipulable NeRF and also proposes a promising method, I recommend a decision of acceptance. The authors should carefully address the concerns of the reviewers in the final version, such as the discussion of novelty and memory usage.